# Moisture-enabled self-charging and voltage stabilizing supercapacitor

Lifeng Wang [1,2,3,4], Haiyan Wang[2], Chunxiao Wu[1,2], Jiaxin Bai[4], Tiancheng He[4], Yan Li [1] ✉, Huhu Cheng [2,4,5] ✉ & Liangti Qu[2,4,5] ✉

Supercapacitor is highly demanded in emerging portable electronics, however, which faces frequent charging and inevitable rapid self-discharging of huge inconvenient. Here, we present a flexible moisture-powered supercapacitor (mp-SC) that capable of spontaneously moisture-enabled self-charging and persistently voltage stabilizing. Based on the synergy effect of moisture-induced ions diffusion of inner polyelectrolyte-based moist-electric generator and charges storage ability of inner graphene electrochemical capacitor, this mp-SC demonstrates the self-charged high areal capacitance of 138.3 mF cm$^{-2}$ and ~96.6% voltage maintenance for 120 h. In addition, a large-scale flexible device of 72 mp-SC units connected in series achieves a self-charged 60 V voltage in air, efficiently powering various commercial electronics in practical applications. This work will provide insight into the design self-powered and ultra-long term stable supercapacitors and other energy storage devices.

Flexible and miniaturized supercapacitors with high power density, long cycling life, and excellent safety are highly demanded in emerging portable electronics of micro aerial vehicles, intelligent robots, human-computer interaction, and the Internet of Things sensing[1–5]. The frequent charging process and inevitable self-discharging of current supercapacitors are dramatically inhibiting the practical convenience of power source devices[6].

Harvesting power from the ambient environment in the highly integrated energy conversion and storage system has become a promising strategy to solve the shortcoming of supercapacitors above mentioned, which can be continuously self-charging, avoiding frequent power source replacement or bulky external charging dependence[7–9]. Ambient solar energy, mechanical movements, and thermal difference have been employed to achieve the electricity generation and storage system by integrating solar cells, piezo/tribo-electric generators, and thermoelectric devices with supercapacitors[7,9–11]. The development of self-charging integrated devices across one-dimensional fibers[12,13], two-dimensional films[9],

three-dimensional bulk structures[14], and textile forms[15,16] has emerged for various applications including health monitoring bioelectronics[17], sensors[18], and wearable electronics[7], which always require external mechanical stimuli or specific geographic and climatic conditions. A recently developed moist-electric generator (MEG) is able to produce electricity by utilizing atmosphere water within the stereoscopic space around us, which could provide a sustainable self-powered strategy in an all-weather and 24-h way[19–24]. It becomes possible to develop the highly integrated energy conversion and storage system without intermittent environmental conditions limitations.

In this regard, the flexible moisture-powered supercapacitor (mp-SC) has been developed, which can be spontaneously self-charging and voltage self-stabilizing when absorbing water from the air. Through layer-by-layer highly-integrating polyelectrolyte-based MEG for electricity generation and graphene electrochemical capacitor (EC) for energy storage, this mp-SC delivers a voltage output of ~0.9 V in 90% relative humidity (RH) air. Especially, the electricity generation provides the constant moist-electric potential that counteracts the

[1]School of Materials Science and Engineering, University of Science and Technology Beijing, Beijing, PR China. [2]Key Laboratory of Organic Optoelectronics & Molecular Engineering, Ministry of Education, Department of Chemistry, Tsinghua University, Beijing 100084, China. [3]State Key Laboratory of Transient Optics and Photonics, Xi'an Institute of Optics and Precision Mechanics, Chinese Academy of Sciences, Xi'an, PR China. [4]State Key Laboratory of Tribology in Advanced Equipment (SKLT), Department of Mechanical Engineering, Tsinghua University, Beijing 100084, China. [5]Laboratory of Flexible Electronics Technology, Tsinghua University, Beijing 100084, PR China. ✉e-mail: liyan2011@ustb.edu.cn; huhucheng@tsinghua.edu.cn; lqu@mail.tsinghua.edu.cn

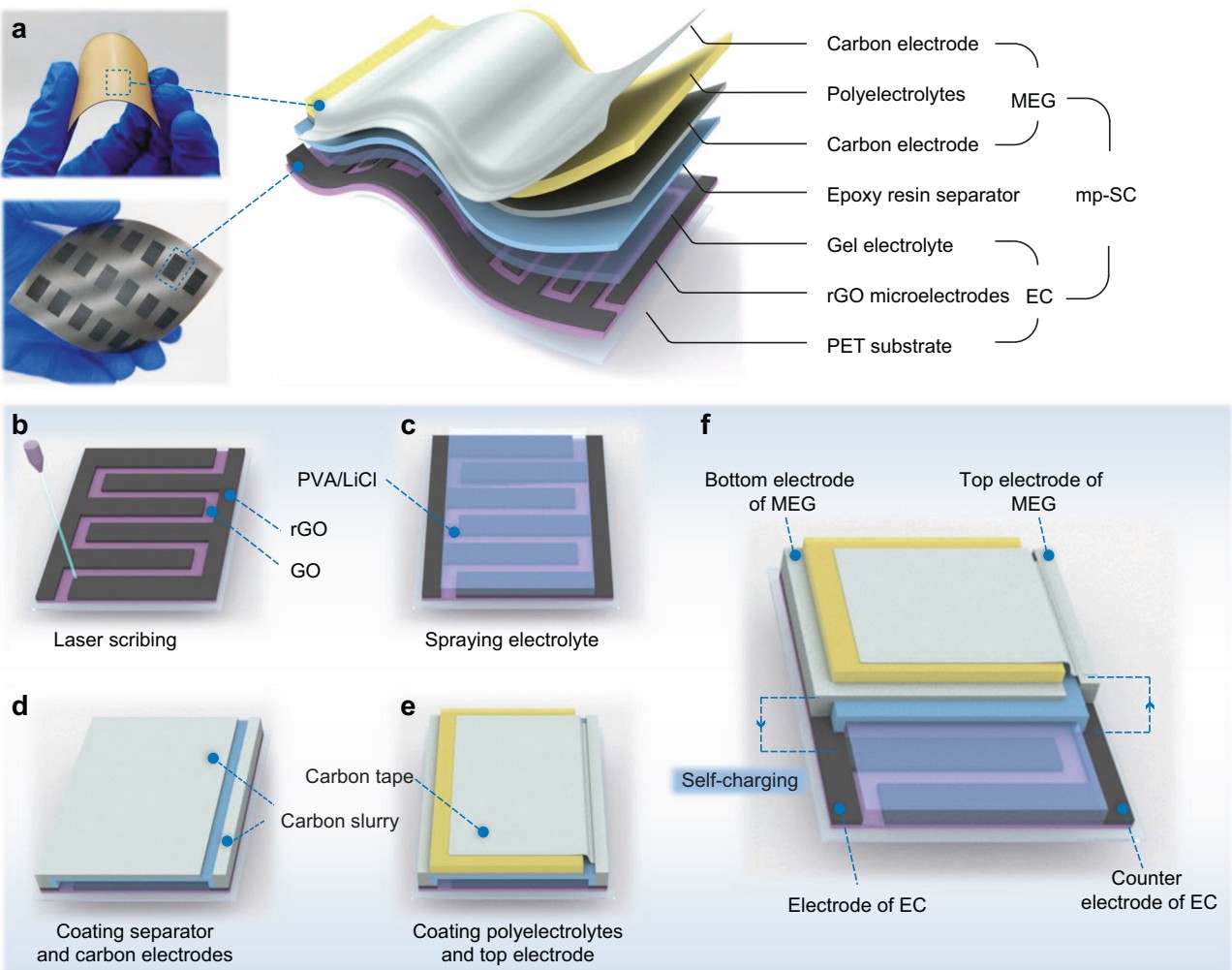

**Fig. 1 | Schematic of the mp-SC. a** Photos of the bilayer polyelectrolytes film and the rGO microelectrodes array obtained by direct-laser writing as well as the scheme of the mp-SC. The mp-SC consists of a polyelectrolyte-based MEG and a graphene EC. **b–e** Schematic illustration of the fabrication process of mp-SC. **f** The schematic diagram of connection between between SC and MEG of the mp-SC.

effect of self-discharge for the electrochemical energy storage, achieving 96.6% voltage maintenance for 120 h without obvious discharge and a high areal capacitance of 138.3 mF cm$^{-2}$ at the current density of 10 μA cm$^{-2}$ of mp-SC. Meanwhile, the power output (49.4 μW cm$^{-2}$) is greatly enhanced by the synergistic effect of electricity generation and stored energy supply that is beyond the individual energy production or storage part. Besides, this mp-SC exhibits remarkable mechanical flexibility and demonstrates ~100% voltage retention under 180° bending for 1000 cycles. The large-scale flexible device array with 72 mp-SC units connected in series reaches 60 V voltage in the air, which powers various electronics such as electronic watches, temperature and humidity meters, and calculators. This study presents a strategy for designing self-powered and ultra-long term stable supercapacitors and paves the way for development of spontaneous energy harvest devices.

## Results

### The configuration and fabrication of mp-SC

The schematic illustration of mp-SC is shown in Fig. 1a. The mp-SC has a multi-layer structure that can be divided into two parts, including graphene-based interdigitated EC and polyelectrolyte-based MEG with rationally designed electrodes. First, a graphene oxide (GO) suspension was blade-coated on a flexible polyethylene terephthalate (PET)

substrate. Direct-laser writing approach was then conducted to construct the pair of interdigitated reduced GO (rGO) microelectrodes on the dried GO film (Fig. 1b). After being coated by the PVA/LiCl electrolyte and the epoxy resin in sequence, the bottom graphene-based interdigitated EC part was constructed (Fig. 1c). Afterward, a conductive carbon paste was coated on top of the epoxy resin and was connected with one rGO microelectrode of EC by the screen-printing process (Fig. 1d). A polyelectrolytes film was adopted and coated on the carbon electrode to harvest energy from moisture (Fig. 1e). Finally, carbon tape is adhered to the polyelectrolytes film and connected to another rGO microelectrode of EC (Fig. 1f), achieving the integration of graphene-based interdigitated EC and polyelectrolyte-based MEG for mp-SC fabrication.

### Characterization and electrochemical performance of the EC part of mp-SC

Figure 2a–c show the black interdigitated rGO microelectrodes and GO intervals of the EC part in mp-SC. The rGO microelectrodes are converted from GO film after laser treatment (355 nm, ~2.2 W) as above mentioned[25]. Due to the photothermal effect, the compact GO film in the interdigitated regions changes into three-dimensional porous rGO microelectrodes (Fig. 2b, c and Supplementary Fig. 1) attached to the GO film[22,26]. Compared with GO, the lower D band in 1350 cm$^{-1}$ of rGO

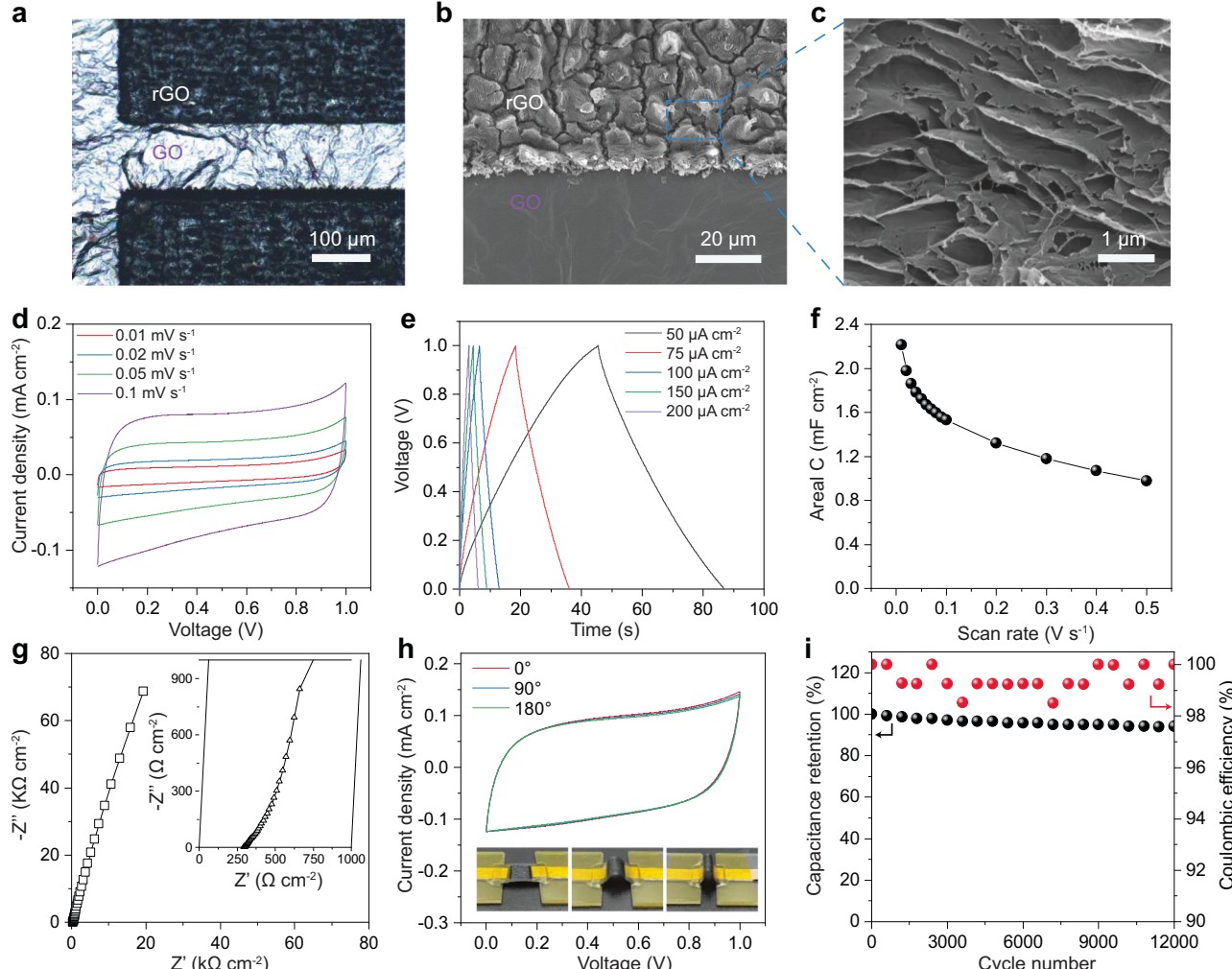

**Fig. 2 | Electrochemical energy storage properties of the EC part of mp-SC.** Photomicrograph (**a**) and SEM images (**b**, **c**) of the laser-reduced rGO microelectrodes. **d** CV curves of the EC part of mp-SC at scan rates of 10 mV s⁻¹, 20 mV s⁻¹, 50 mV s⁻¹, and 100 mV s⁻¹. **e** Areal capacitance versus scan rate of the EC at scan rates between 10 and 500 mV s⁻¹. **f** Galvanostatic charge/discharge curves at different current densities. **g** Electrochemical impedance spectroscopy analysis of the EC part of mp-SC, the inset shows the high-frequency region of the EC. **h** CV curves of the EC part of mp-SC under normal and bent conditions, inserted optical images show the EC at different bending angles. **i** Cycling stability and Coulombic efficiency of the EC after 12,000 charge-discharge cycles.

microelectrodes in Raman spectroscopy (Supplementary Fig. 2a) result indicates the oxygen functional groups and other defects are significantly decreased. X-ray photoelectron spectroscopy (XPS) spectra (Supplementary Fig. 2b, c) reveal the O atom ratio in rGO microelectrodes is about 14.79% that is much lower than that of GO (~31.29%), further resulting in a drastic increase in electrical conductivity from $5.09 \times 10^{-3}$ S m⁻¹ (GO) to $2.73 \times 10^{3}$ S m⁻¹ of rGO microelectrodes (Supplementary Fig. 2d). The porous structure and the high electrical conductivity will facilitate the electrolyte diffusion and electron transport in the microelectrodes[27], endowing them with promising energy storage capabilities.

Cyclic voltammetry (CV) and galvanostatic charge−discharge profiles were recorded at the potential window between 0 and 1 V to evaluate the electrochemical performance of the EC part in mp-SC. The nearly rectangular CV curves (Fig. 2d) indicate the typical double-layer capacitive behavior of the rGO microelectrodes[28,29]. The area-specific capacitance of the EC part according to the CV profiles is shown in Fig. 2e, which shows the areal capacitance of 2.21 mF cm⁻² at a current density of 10 mA cm⁻², comparable to values of the state-of-the-art carbon-based supercapacitors[28]. Figure 2f displays nearly symmetric triangular galvanostatic charge and discharge profiles with fast charge-

discharge characteristics and high Coulombic efficiencies close to 100%. Besides, epoxy encapsulation shows negligible influence on the capacitance performance of the EC (Supplementary Fig. 3). For instance, by connecting ECs in series or in parallel, the devices achieved a further expanded voltage window or boosted capacitance (Supplementary Fig. 4). The nearly vertical slope in the low-frequency region shown in Nyquist plots (Fig. 2g) further confirms their ideal double-layer capacitive behavior[30]. The short 45° line inserted in Fig. 2g proves that the interconnection of the porous structures exists in the rGO film by the de Levie model[31]. Furthermore, the EC part is highly robust and flexible shown in Fig. 2h. According to the CV curves, almost 100% of initial capacitance is maintained even under a bending angle of 180°. Besides, the EC demonstrates an impressive capacitance retention of 92.5% and a high Coulombic efficiency of 99% after more than 120,000 charge-discharge cycles (Fig. 2i).

**The power generation performance of the MEG part of mp-SC**
For the MEG part of mp-SC, polydiallyl dimethyl ammonium chloride (PDDA) and polystyrene sulfonic acid (PSS) bilayer polyelectrolytes film are adopted (Fig. 3a). Figure 3b shows the cross-section scanning electron microscopy (SEM) image of this bilayer polyelectrolytes film,

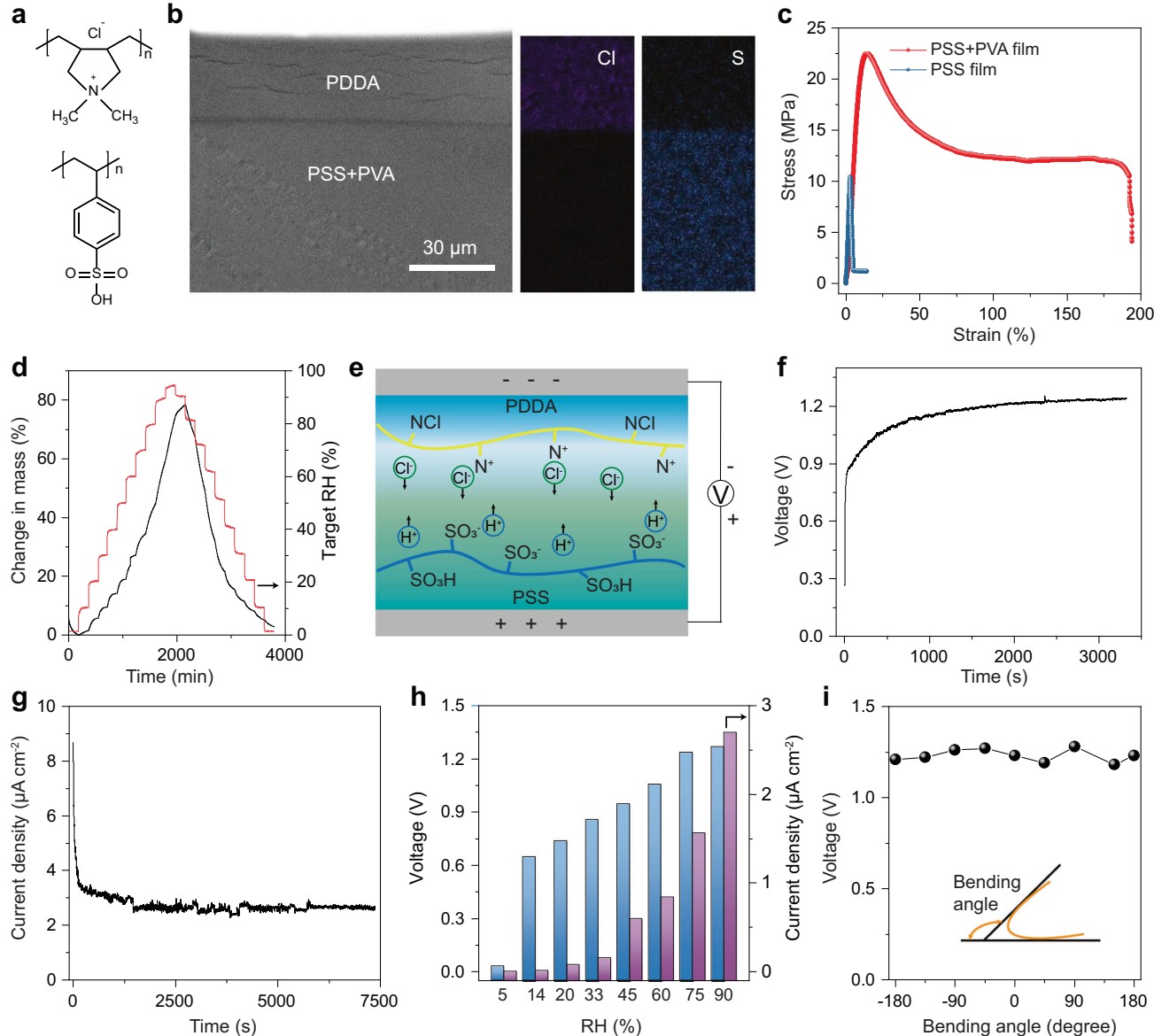

**Fig. 3 | Moisture energy-harvesting properties. a** Chemical structures of the PSS and PDDA. **b** Cross-section SEM image of the polyelectrolytes film and corresponding energy-dispersive X-ray spectroscopy (EDS) mappings of Cl and S. **c** The stress-strain curve of the polyelectrolytes film. **d** The mass change of the polyelectrolytes film at different RH. **e** Schematic illustration of the polyelectrolyte-based MEG. The output voltage (**f**) and current density (**g**) of the MEG. **h** The output voltage and current density of the MEG under different RH from ~5 to ~95%. **i** Voltage retention of the generator under different bending angles.

demonstrating the PDDA layer and PSS layer clearly. Cl and S elements mapping via energy dispersive spectroscopy (EDS) (Fig. 3b) further confirm the successful fabrication of bilayer structure because the Cl element is mainly distributed at the PDDA layer and the S element is mainly distributed at the PSS layer. In the PSS layer, PVA is incorporated into the polyelectrolytes film with a mass ratio of 14.5% that could improve the mechanical properties and flexibility (Fig. 3c). Simultaneously, the PVA has contributed a fraction of dissociable hydrogen ions. The failure strain of the composite film is 4.5 times higher than pure PSS film as shown in Fig. 3c. Owing to the abundant -SO₃H, -OH, and -NCl functional groups content (Supplementary Fig. 5), the polyelectrolytes film could spontaneously absorb or desorb water when the ambient humidity changes and a high water absorption capacity of 78.3% is achieved at 95% RH and 25 °C (Fig. 3d). After sandwiching the bilayer polyelectrolytes film between the blade-coated carbon bottom electrode (Supplementary Fig. 6) and carbon tape top electrode, the MEG part in mp-SC is obtained.

The power generation mechanisms are investigated by simulations and experimental studies (Supplementary Figs. 7 and 8). After exposure to moisture, polyelectrolytes film harvests water molecules from the ambient environment and dissociates movable $H^+$ and $Cl^-$. The unevenly distributed ions will create an electric potential difference between the two electrodes driven by a concentration gradient (Fig. 3e). The surface potential values of −0.87 V for PSS and 0.92 V for PDDA have been conclusively confirmed via Kelvin probe force microscopy tests. These potential disparities arise from the gradient distribution of mobile $H^+$ and $Cl^-$ ions. The electric power generation process is demonstrated using Poisson and Nernst−Planck (PNP) theory (Supplementary Fig. 7). A 1.07 V open-circuit voltage is attained through the gradual diffusion of $H^+$ and $Cl^-$ ions within the bilayer polyelectrolytes film, closely aligning with the experimental test results.

The power generation performance of the MEG is then tested as shown in Supplementary Fig. 9. After adsorbing water from the air

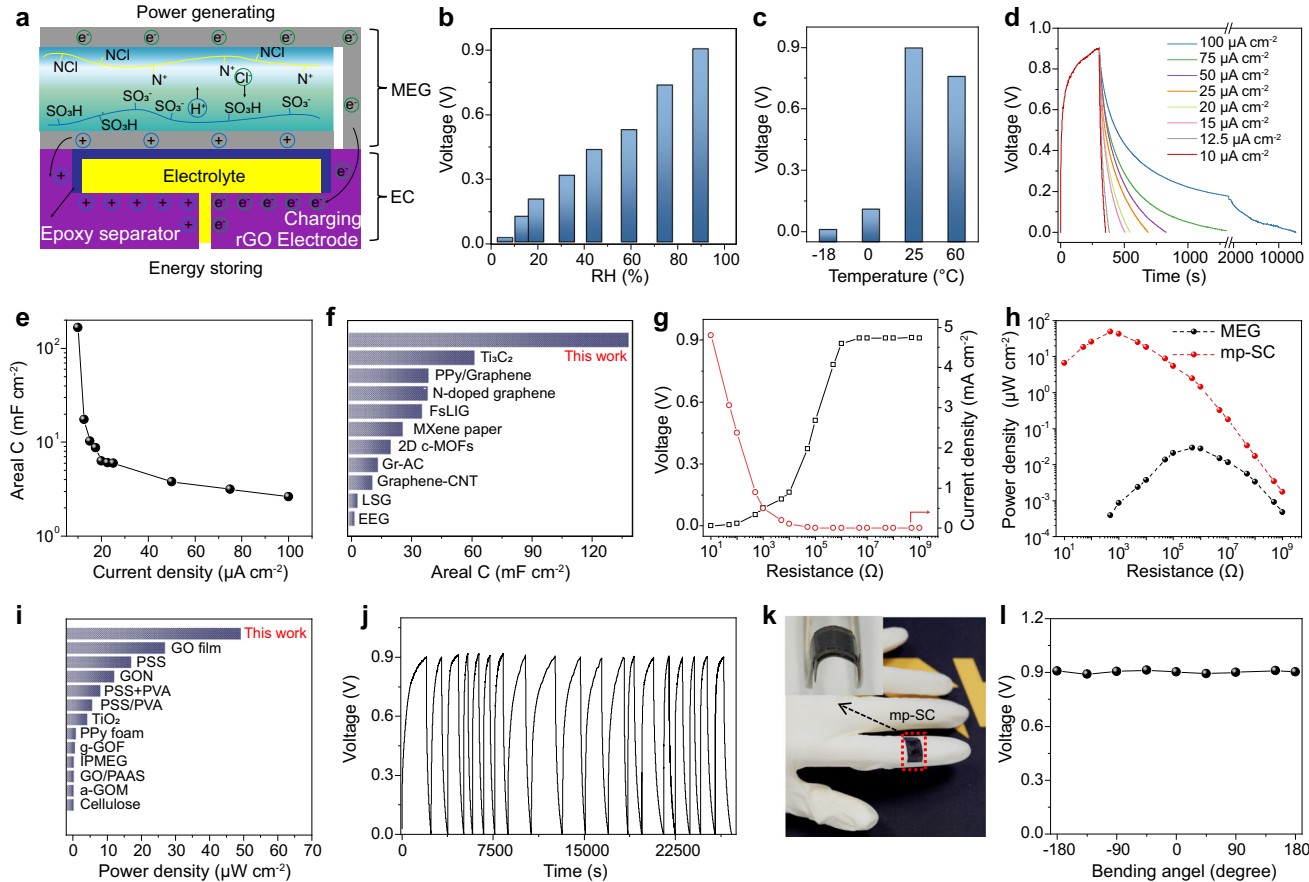

**Fig. 4 | The performance of the mp-SC. a** Schematic of the charging process of the mp-SC. **b** The voltage output of the mp-SC at different RH. **c** The voltage output of the mp-SC at different temperature. **d** Galvanostatic charge-discharge profiles of the mp-SC at different current densities. **e** The areal capacitance of the mp-SC at different current densities. **f** Comparison of the capacitance of the mp-SC with the recently reported ECs. Output voltage and current (**g**) as well as the power density (**h**) of the mp-SC with different electric resistors. **i** Comparison of the power density of the mp-SC and the recently reported moisture power generators. **j** The moisture-charging/discharging cycling stability of the mp-SC at 90% RH. **k** Optical images of the mp-SC at the bending state. **l** The voltage output of the device at different bending states. The voltage retention rate of the mp-SC after cyclic bending and flat test.

(90% RH and 25 °C), a sustained voltage of 1.27 V and a current density of 2.7 µA cm$^{-2}$ are generated as shown in Fig. 3f, g. Figure 3h displays the electricity generation of the MEG part in mp-SC at different humidity air. Even at very low humidity of 15%, this MEG part has a favorable voltage of 0.65 V, revealing excellent environmental suitability. With the rise in RH, the produced voltage and current increases because of the enhanced ion dissociation ratio and ion migration rate of polyelectrolytes related to the higher water content ratio[32–34]. Meanwhile, the MEG part in mp-SC exhibits a stable voltage output under the bending state, indicating excellent mechanical flexible properties (Fig. 3i).

**Moisture-powered energy storage performance of mp-SC**
By integrating the energy generation part and energy storage part with well-designed electrodes as indicated in Fig. 4a and Supplementary Fig. S10, this mp-SC can absorb water from the air and gradually dissociates and releases moveable H$^+$ and Cl$^-$ at the MEG part. Then, the asymmetrically distributed migratable ions (H$^+$ and Cl$^-$) in the bilayer polyelectrolytes film will diffuse to the opposite side driven by the concentration difference above mentioned and produce electrical power. The voltage generated by the polyelectrolytes film is applied the carbon electrodes and then acts on the rGO microelectrodes of the EC through the internally connected electrodes. Under the applied voltage bias, the electrolyte within the EC selectively adsorbs onto the surface of the porous electrodes, facilitating the conversion of

electrical energy into chemical energy. Based on this method, the power energy produced by the MEG is storaged in the EC via the electric double-layer mechanism[2]. Furthermore, the mp-SC is capable to harvest energy in a wide RH or temperature range (Fig. 4b, c; Supplementary Fig. S11, Video S1, and Video S2) indicating its robustness toward harsh environmental conditions. As shown in Fig. 4d, the voltage of mp-SC can spontaneously self-charged to ~0.9 V after 380 s when absorbing water in 90% RH air and the stored electricity can be discharged at constant current density for power output. As a new-type moisture-powered energy storage device, this mp-SC represents distinctive moisture-induced self-charging and electricity-discharging curves. The moisture-induced self-charging voltage curve experiences a rapid initial increase, followed by a gradual stabilization, which is determined by the moisture-enabled energy generation process. Besides, the mp-SC represents a typical galvanostatic discharge behavior at different current densities. Notably, the discharge time is about 12,985 s at a current density of 10 µA cm$^{-2}$, which is much longer than that of individual EC parts (459 s) because of the synergistic effect of electricity generation and stored energy release. As a result, the mp-SC exhibits an ultra-high areal capacitance of 138.3 mF cm$^{-2}$. The areal capacitance is one or two orders of magnitude greater than previously reported graphene supercapacitors and some pseudocapacitors (Fig. 4e, f)[26,28,35–37].

Meanwhile, the output voltage increases and the output current decreases as the electric load varied from 10 Ω to 1 GΩ (Fig. 4g),

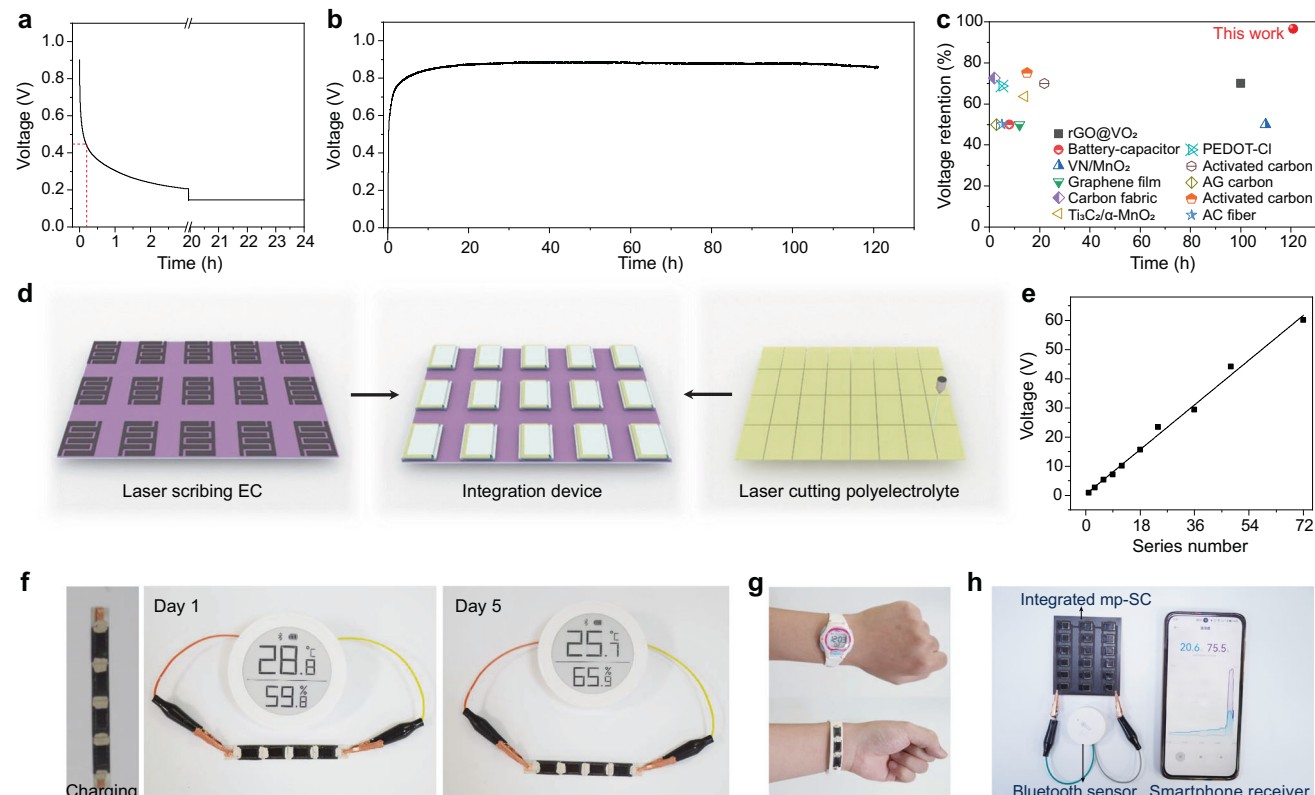

**Fig. 5 | Applications of the mp-SC. a** The self-discharge curve of the individual EC. **b** The continuous voltage output of the mp-SC for over 120 h at 90% RH and 25 °C. **c** Comparison of voltage retention rate of the mp-SC and the reported ECs. **d** Schematic diagram of large-scale fabrication of mp-SC. **e** A plot of voltage output with different serial numbers of mp-SC. **f** The self-charging process of the mp-SCs and a demonstration of a commercial temperature-humidity monitor are directly driven by four mp-SCs connected in serial as well as the monitor is driving again 5 days later. **g** Demonstration of three mp-SCs coherent in the wristband and serving as the power supply for an electronic watch. **h** Demonstration of applying the large-scale integrated mp-SCs for powering a commercial thermohygrometer and transmitting the data to a smartphone through Bluetooth.

achieving a maximum value of ~49.4 µW cm⁻² at an optimal resistance of about 500 Ω. Generally, high power density is difficult to achieve for MEG due to its relatively high internal resistance. In combination with high energy density EC, the maximum power density is 1500 times higher than the polyelectrolyte-based generator and outperforms the state-of-the-art moisture electric power generators as shown in Supplementary Table 1 and Fig. 4h, i[21,22,38-46]. Figure 4j represents the moisture-charging/discharging cycling performance with the same discharge current density (50 µA cm⁻²) under high RH (~90% RH at 25 °C). The mp-SC maintains durable moisture-charging/discharging output performance more than 15 cycles. Moreover, the integrated device demonstrates a good voltage retention rate of ~100% under bending and flat states after 1000 bending cycles, showing the remarkable mechanical durability (Fig. 4k, l, Supplementary Fig. 12).

Compared with individual supercapacitor energy storage (Fig. 5a), this mp-SC self-powered energy storage device prevents the significant voltage drop. Figure 5b represents that mp-SC has an ultra-long open-circuit voltage maintenance time of up to 120 h. While the self-discharge curves of individual EC show the open-circuit voltage drops to 0.45 V sharply within 10 min due to the intrinsic self-discharge effect of supercapacitors[47]. This can be contributed to that the polyelectrolyte-based generator continuously charges the energy storage part in mp-SC, making the unique voltage retention behavior. The voltage retention rate of the mp-SC is up to 96.6% after 120 h, which is superior to those of supercapacitors by suppressing the self-discharging processes (Fig. 5c) through modifying the electrode, introducing additives or ion-exchange membranes and chemically active separation materials[48-55].

The large-scale integration of the device is critical to supply sufficient power for electronics. Herein, the laser processing, screen printing, and spraying processes are easy to scale up to obtain the integration of mp-SC units as shown in Fig. 5d. By connecting in series, the voltage increases linearly with the number of the mp-SCs, allowing for a maximum output voltage of ~60 V with 72 units just putting them in the air (90% RH and 25 °C). Four mp-SCs are connected in series to harvest and store energy from the ambient environment. Subsequently, these devices can be utilized to power a commercial temperature-humidity monitor (~60% RH at 25 °C). To demonstrate stability, the temperature-humidity monitor can be driven again by the mp-SCs five days later due to its robust and stable output performance. Furthermore, the flexibility of the mp-SC enable it to be seamlessly integrated into the wristband of a conventional electronic watch and powering the watch for minutes after self-charging even under bending or twisting conditions (Fig. 5g and Video S3). In addition, four mp-SCs connect in series are able to directly power an electronic calculator in ambient conditions (Supplementary Fig. 13 and Video S4) without requiring supplementary power management circuits. This impressive capability highlights the potential of mp-SC as a promising long-term self-powered source for wearable and flexible electronics. Moreover, the integrated mp-SCs (Fig. 5h) can directly power a Bluetooth-enabled sensor and transmit the data to a smartphone through Bluetooth, demonstrating their immense potential for powering distributed Internet of Things (IoT) sensors.

## Discussion

To summarize, this study has developed a flexible, durable, and ultra-stable flexible mp-SC. By combining polyelectrolyte-based MEG for

electricity generation with graphene EC for energy storage, the mp-SC can harvest energy directly from atmospheric moisture, generate electric power, and store it simultaneously. This mp-SC delivers a voltage output of ~0.9 V in 90% RH air 96.6% and has excellent voltage maintenance for 120 h. The high areal capacitance of 138.3 mF cm$^{-2}$ at the current density of 10 μA cm$^{-2}$ and the power output (49.4 μW cm$^{-2}$) of mp-SC is achieved based on the synergistic effect of electricity generation and stored energy supply. The large-scale flexible mp-SC units connected in series can power various electronics such as electronic watches, temperature and humidity meters, and calculators, which delivers great potential for the development of self-powered and ultra-long term stable supercapacitors for future applications.

## Methods

### Materials
GO dispersion was prepared by using a modified Hummers' method[2,55-57]. PSS (with weight-averaged molecular weight, $M_w = 75,000$, 30 wt% in water), PVA ($M_w = 15,000$), PDDA ($M_w = 100,000$, 35 wt% in water) were purchased from Shanghai Aladdin Biochemical Technology Co., Ltd and were used as received without further purification. Conductive carbon paste (CH-8) was purchased from Jelcon Corp., Japan. Conductive carbon tape (No. 7321) was purchased from Nisshin EM Co.

### Preparation of bilayer polyelectrolytes film
40 g PVA was added into 360 g of deionized water, and the mixture was stirred at 90 °C in a water bath until the PVA is completely dissolved, resulting in a 10 wt% PVA solution. 2 g PVA solution was added in 3.92 g of 30 wt% PSS solution and 4 g deionized water, and was stirred at room temperature to obtain a homogeneous solution. Adding the resulting solution to a laboratory dish with a diameter of 9 cm, and drying 35 °C and 50% RH to obtain a PVA + PSS composite film. Next, 1.5 mL of PDDA was sprayed onto the bilayer film using a pressure of 50 Psi. Hot air is used to dry the sprayed PDDA during the spraying process to prevent it from dissolving into the film. SEM tests indicate the thickness of the bilayer film was ~100 μm and the ratio of PDDA: PSS + PVA was 3:7. After the preparation process, the film was cut into small pieces (6.6 × 6.6) mm using a 355 nm nanosecond laser (3 W) with a pulse repetition frequency of 20 kHz and scanning speed of 20 mm s$^{-1}$.

### The fabrication process of the mp-SC
The integrated device was prepared by layer-by-layer stacking processing. First, a GO (13 mg mL$^{-1}$) dispersion was coated onto a PET substrate, with a height of about 1.5 mm and dried in a 40 °C oven. The obtained GO film with a thickness of about 8 μm tightly loaded onto the flexible PET substrate. The rGO microelectrodes were fabricated using direct-laser writing method (~2.2 W, 355 nm, 93 kHz), with a size of 33.8 mm$^2$ for the electrode and the gap area. The laser scanning speed and pulse duration were 500 mm s$^{-1}$ and 9.7 μs, respectively. The PVA/LiCl gel electrolyte (5 M) was then sprayed onto the microelectrode area with a mask using an airbrush, with the pressure set to 50 psi[2]. A single-component insulating epoxy resin was used to print as a protective layer in the electrolyte area by screen printing and then cured at 60 °C for 6 h in an oven. In addition, a conductive carbon paste layer (pre-doped with 1% carbon fiber to enhance the mechanical properties of the carbon electrode) was printed to connect one of the current collectors of the EC as serve as the bottom electrode of the MEG. A bilayer polyelectrolytes film was coated on the bottom electrode. Finally, a conductive carbon tape adhered to the polyelectrolytes film serve as the top electrode of MEG and connected to the other electrode of the EC. The mp-SC was successfully fabricated.

### Characterization and measurement
Optical images were captured using the Axio Scope A1 optical microscope by Zeiss (Germany). SEM images were taken with a Sirion-200 field-emission scanning electron microscope made by FEI (USA), and EDS was performed using an INCA Energy EDS system by Oxford Instruments (UK). Raman spectra were detected using a LabRAM HR Raman spectrometer (Horiba Jobin Yvon, Japan) equipped with a 532 nm laser. XPS data were acquired using an ESCALAB 250Xi X-ray photoelectron spectrometer by Thermo Fisher Scientific (USA). Electronic conductivity was determined by the I-V test, with the formula $\sigma = (I/V) \times (l/A)$, where $A$ is the cross-sectional area of the test sample, and $l$ is the distance between the two electrodes. Stress–strain curves were obtained using a tensile machine (Instron 5943) with a constant strain rate of 1 mm min$^{-1}$. Fourier transform infrared spectra were obtained using a Spectrum Two infrared spectrometer by PerkinElmer (USA). Dynamic vapor sorption (DVS) measurements were performed using a DVS-1000 dynamic vapor sorption analyzer (Surface Measurements Systems, UK). A weight change of less than 0.002% was considered to reach the sorption/desorption equilibrium state. Water contact angles were measured using an OCA 15 contact angle tester, while the surface potential was measured with an Asylum Research Cypher ES Kelvin probe microscope (Oxford Instruments, UK).

The electrochemical performance of the EC was tested by an electrochemical workstation (CHI 760E, Shanghai Chenhua, China). Two-electrode systems were adopted for CV, constant current charge-discharge, and electrochemical impedance spectroscopy (EIS) testing. The area capacity calculation was based on the sum of the area occupied by the finger electrodes and the area between them. The voltage window was set at 1 V. The frequency range for EIS testing was from 0.01 Hz to 100 kHz, and the bias voltage was set at 5 mV. The electrochemical energy storage tests were carried out in a sealed container at 60% RH and 25 °C.

The capacitance of the EC was calculated from the CV curves using the formula:

$$C = \frac{\int_{V_i}^{V_f} I(V) dV}{\nu * \Delta V} \tag{1}$$

$$C_A = \frac{C}{A_{device}} \tag{2}$$

Where $C$ is the capacitance of the EC (F), $\nu$ is the scan rate (V s$^{-1}$), $I$ is the current response (A), $V_i$ and $V_f$ are the initial and final voltages (V), and $\Delta V$ (V) is the voltage window. $A_{device}$ (cm$^2$) is the sum of the area occupied by the electrodes and the area between the electrodes. $C_A$ (F cm$^{-2}$) is the areal capacitance of the EC.

The voltage and current of MEG and mp-SC were measured using a Keithley 2612 multimeter, which was controlled by a LabView-based data acquisition system. The open-circuit voltage testing mode was used with a current bias of 0 nA. For current measurement, the short circuit current testing mode was used with a voltage bias of 0 V.

### Reporting summary
Further information on research design is available in the Nature Portfolio Reporting Summary linked to this article.

## Data availability
The raw data generated in this study are provided in the Supplementary Information. All data are available from the corresponding author upon request. Source data are provided with this paper.

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

## Acknowledgements

This work was supported by the National Natural Science Foundation of China (No.22035005, L.Q., 52350362, 52022051, 22075165, 52090032, H.C., 52073159, L.Q., 22175019, Y.L.), Tsinghua-Foshan Innovation Special Fund (2020THFS0501, L.Q.).

## Author contributions

L.Q., H.C., and Y.L. proposed and supervised this project. L.W. and H.C. designed the experiments and accomplished the original draft. H.W., C.W., J.B., and T.H. contributed to material preparation and the experimental setup. All authors contributed to writing and reviewing the manuscript.

## Competing interests

The authors declare no competing interests.
