## [Peer Review File · Nature Communications]

Moisture-enabled self-charging and voltage stabilizing supercapacitorREVIEWER COMMENTS

Reviewer #1 (Remarks to the Author):

In this manuscript, the authors report a very interesting flexible moisture-powered supercapacitor (mp-SC) that capable of spontaneously self-charged by absorbing water from the air. This device is able to deliver a voltage output reach ~ 0.9 V in 90% RH air with a high areal capacitance. Moreover, they show the potential applications using the integrated mp-SCs. This work is very innovative, and cleverly uses the mp-SC design to solve the two shortcomings of supercapacitors (facing frequent charging and inevitable rapid self-discharging). In my opinion, this excellent result is definitely worthy of publication in Nature Communications. Of course, before its publication, the following questions need to be well answered and revised.

1. Why can the voltage of a SC device reach 1V, but that of a mp-SC can only reach 0.9 V?
2. We all well know that the change of ambient temperature has a decisive impact on the evaporation of water inside a device, thereby affecting the overall performance of the device. The related research in this paper is relatively simple. I strongly recommend supplementing the corresponding studies and experiments, and adding the results and graphs to the main text.
3. The authors mentioned in the text that "The mp-SC maintains outstanding cycling stability after 15 cycles, demonstrating stable and durable output performance". However, they also said in the abstract and text that "this mp-SC demonstrates $\sim 100\%$ performance retention under bending for 1000 cycles", and "the integrated device demonstrates a good capacitance retention rate of $\sim 100\%$ under bending states". Whether the two statements are contradictory each other?
4. In Figure 4, most of the graphs most have incorrect title numbers. Please revise carefully.
5. In addition, I suggest the authors to appropriately reduce the references that are their own works.

Reviewer #2 (Remarks to the Author):

This manuscript reported on the development of an interesting flexible moisture-powered supercapacitor (mp-SC), which can absorb water from air and then self-charge electricity spontaneously. This is a wonderful experiment and the novel device is well-designed. The attractive power generation performance, ultra-long term energy storage ability, remarkable mechanical flexibility and large-scale integration capacity highlights the potential of mp-SCs as brand-new self-powered sources for wearable and flexible electronics. The study is of high quality and I strongly suggest considering this work for publications in Nature Communications after addressing some minor concerns as listed below:

1. It is not clear what the carbon electrode is and how it looks like after blade coating. Please supplement more information.
2. The authors should consider providing additional discussions on existing self-powered supercapacitors in the introduction section.
3. How about the thickness of this flexible mp-SC?
4. The scale bars in Figure 3b and currently missing from the manuscript.
5. In Figure 4i, the authors compared the performance with previously reported moist-electric generators. However, given the abundance of existing reports on moist-electric generators, the authors should provide a more detailed performance comparison, including materials, device structures, and other relevant factors.
6. In Figure 5h, it is not explicitly stated whether the thermohygrometer directly transmits the data to a smartphone through Bluetooth or if any relay device is involved in the process.

Reviewer #3 (Remarks to the Author):

The author presents the novel configuration of a moisture-based nanogenerator and supercapacitor. Overall, the manuscript is well-written, but the current manuscript should be considered after addressing the following major questions.

1. The title of the manuscript seems too broad. It should clearly describe the novelty of the work instead of the "review article"-like title.
2. The overall configuration should be reviewed. The author should explain the stacked structure of the supercapacitor and mp-SC more systematically. For better understanding, it is recommended to show the stacked structure of mp-SC in Figure A with well-defined connections between SC and MEG. For example, in Figures 1f and 1g, it is quite difficult to understand, how the electrodes are connected to Supercapacitor.
3. Also, the energy harvesting mechanism and energy storage from the moisture should be more clearly depicted as a schematic. In the current manuscript, it is too hard to understand the energy harvesting mechanism. Clearer schematic as well as description are required.
4. The author should show the video for real time generation of the electricity from the moisture-environment. For example, in real time, continuous changing environment from dry state to humid state with generation of electricity in oscilloscope would be enough.
5. The author should explain the different parameters in electrochemical impedance spectroscopy of supercapacitors i.e. charge transfer resistance, and series resistance under the different bending conditions. Also, the strong connection between the charge transfer resistance and charging time when connected with MEG should explained in detail. The EIS part at the higher frequency region should be present in the inset of Figure 2g
6. In general, the fabricated supercapacitor lacks novelty in terms of material as rGO has already reported multiple times.
7. It is commonly observed that potential drops if the supercapacitor is connected without any rectifier. The author should report the potential drops with different levels of humidity level once the supercapacitor is fully charged. How does the author balance the charge flow or overcome the reverse flow of current from the supercapacitor to MP. The reverse flow can again neutralize the charge distribution of H⁺ and Cl⁻. Please explain the mechanism in detail
8. The author should explain the detailed mechanism that how functional groups in the bipolar layer can lead to absorbing and desorbing water.
9. The author should explain the optimization of thickness for PSS and PDDA. The effect of surface potential on the thickness of PSS and PDDA should be addressed in detailed
10. The captions in Figure (4) are not well matched with the figure number. Please correct it.

REVIEWER COMMENTS

Reviewer #1 (Remarks to the Author):

In this manuscript, the authors report a very interesting flexible moisture-powered supercapacitor (mp-SC) that capable of spontaneously self-charged by absorbing water from the air. This device is able to deliver a voltage output reach ~ 0.9 V in 90% RH air with a high areal capacitance. Moreover, they show the potential applications using the integrated mp-SCs. This work is very innovative, and cleverly uses the mp-SC design to solve the two shortcomings of supercapacitors (facing frequent charging and inevitable rapid self-discharging). In my opinion, this excellent result is definitely worthy of publication in Nature Communications. Of course, before its publication, the following questions need to be well answered and revised.

1. Why can the voltage of a SC device reach 1V, but that of a mp-SC can only reach 0.9 V?
2. We all well know that the change of ambient temperature has a decisive impact on the evaporation of water inside a device, thereby affecting the overall performance of the device. The related research in this paper is relatively simple. I strongly recommend supplementing the corresponding studies and experiments, and adding the results and graphs to the main text.
3. The authors mentioned in the text that “The mp-SC maintains outstanding cycling stability after 15 cycles, demonstrating stable and durable output performance”. However, they also said in the abstract and text that “this mp-SC demonstrates $\sim 100\%$ performance retention under bending for 1000 cycles”, and “the integrated device demonstrates a good capacitance retention rate of $\sim 100\%$ under bending states”. Whether the two statements are contradictory each other?
4. In Figure 4, most of the graphs most have incorrect title numbers. Please revise carefully.
5. In addition, I suggest the authors to appropriately reduce the references that are their own works.

Reviewer #2 (Remarks to the Author):

This manuscript reported on the development of an interesting flexible moisture-powered supercapacitor (mp-SC), which can absorb water from air and then self-charge electricity spontaneously. This is a wonderful experiment and the novel device is well-designed. The attractive power generation performance, ultra-long term energy

storage ability, remarkable mechanical flexibility and large-scale integration capacity highlights the potential of mp-SCs as brand-new self-powered sources for wearable and flexible electronics. The study is of high quality and I strongly suggest considering this work for publications in Nature Communications after addressing some minor concerns as listed below:

1. It is not clear what the carbon electrode is and how it looks like after blade coating. Please supplement more information.
2. The authors should consider providing additional discussions on existing self-powered supercapacitors in the introduction section.
3. How about the thickness of this flexible mp-SC?
4. The scale bars in Figure 3b and currently missing from the manuscript.
5. In Figure 4i, the authors compared the performance with previously reported moist-electric generators. However, given the abundance of existing reports on moist-electric generators, the authors should provide a more detailed performance comparison, including materials, device structures, and other relevant factors.
6. In Figure 5h, it is not explicitly stated whether the thermohygrometer directly transmits the data to a smartphone through Bluetooth or if any relay device is involved in the process.

Reviewer #3 (Remarks to the Author):

The author presents the novel configuration of a moisture-based nanogenerator and supercapacitor. Overall, the manuscript is well-written, but the current manuscript should be considered after addressing the following major questions.

1. The title of the manuscript seems too broad. It should clearly describe the novelty of the work instead of the “review article”-like title.
2. The overall configuration should be reviewed. The author should explain the stacked structure of the supercapacitor and mp-SC more systematically. For better understanding, it is recommended to show the stacked structure of mp-SC in Figure A with well-defined connections between SC and MEG. For example, in Figures 1f and 1g, it is quite difficult to understand, how the electrodes are connected to Supercapacitor.
3. Also, the energy harvesting mechanism and energy storage from the moisture should be more clearly depicted as a schematic. In the current manuscript, it is too hard to understand the energy harvesting mechanism. Clearer schematic as well as description are required.
4. The author should show the video for real time generation of the electricity from

the moisture-environment. For example, in real time, continuous changing environment from dry state to humid state with generation of electricity in oscilloscope would be enough.

5. The author should explain the different parameters in electrochemical impedance spectroscopy of supercapacitors i.e. charge transfer resistance, and series resistance under the different bending conditions. Also, the strong connection between the charge transfer resistance and charging time when connected with MEG should be explained in detail. The EIS part at the higher frequency region should be present in the inset of Figure 2g

6. In general, the fabricated supercapacitor lacks novelty in terms of material as rGO has already reported multiple times.

7. It is commonly observed that potential drops if the supercapacitor is connected without any rectifier. The author should report the potential drops with different levels of humidity level once the supercapacitor is fully charged. How does the author balance the charge flow or overcome the reverse flow of current from the supercapacitor to MP. The reverse flow can again neutralize the charge distribution of H^+ and Cl^- . Please explain the mechanism in detail

8. The author should explain the detailed mechanism that how functional groups in the bipolar layer can lead to absorbing and desorbing water.

9. The author should explain the optimization of thickness for PSS and PDDA. The effect of surface potential on the thickness of PSS and PDDA should be addressed in detailed

10. The captions in Figure (4) are not well matched with the figure number. Please correct it.

RESPONSE TO REVIEWERS' COMMENTS

Reviewer: 1

Comments:

In this manuscript, the authors report a very interesting flexible moisture-powered supercapacitor (mp-SC) that capable of spontaneously self-charged by absorbing water from the air. This device is able to deliver a voltage output reach ~ 0.9 V in 90% RH air with a high areal capacitance. Moreover, they show the potential applications using the integrated mp-SCs. This work is very innovative, and cleverly uses the mp-SC design to solve the two shortcomings of supercapacitors (facing frequent charging and inevitable rapid self-discharging). In my opinion, this excellent result is definitely worthy of publication in Nature Communications. Of course, before its publication, the following questions need to be well answered and revised.

Response: We thank the reviewer for giving positive comment and insightful advice to help us improve the manuscript.

1. Why can the voltage of a SC device reach 1V, but that of a mp-SC can only reach 0.9 V?

Response: Indeed, the voltage window of the individual electrochemical capacitor (EC) ranges from 0 to 1 V, as indicated by the cyclic voltammetry profiles (Fig. R1a). Meanwhile, the individual polyelectrolyte-based MEG demonstrates a sustained output voltage of 1.27 V at 90% relative humidity and 25°C (Fig. R1b). For mp-SC, the final voltage could be determined by the balance between moisture induced charging current of MEG and leakage current of EC inner the whole device. Because the phenomenon of open circuit self-discharge is inevitable in inner EC. The charge storage process entails unstable electrostatic interactions, as ions absorbed in the electrodes possess higher Gibbs free energy compared to their discharged state (Energy Environ. Sci., 2021,14, 2859; J. Am. Chem. Soc. 2017, 139, 9985). This leads to the potential decay process and leakage current. As depicted in Fig. R1c, the open-circuit voltage of the EC sharply drops to 0.45 V within 10 minutes because of the leakage current induced the self-discharge. Moreover, the leakage current is related to the open circuit voltage. As illustrated in Figure R1d, the leakage current gradually increases as the open circuit voltage of the electrochemical capacitor increases from 0.5 to 1 V. At the same time, the MEG demonstrates a stable charging current of $2.7 \mu\text{A cm}^{-2}$. Therefore, the output voltage of the final mp-SC reaches a stable value of about 0.9 V (Fig. R1e) when the leakage current of inner EC balances with charging current of inner MEG induced by moisture. The related discussion has been added in the revised manuscript for the explanation of the device performance.

Fig. R1 **a** CV curves of the EC part of mp-SC at scan rates of 10 mV s^{-1} , 20 mV s^{-1} , 50 mV s^{-1} and 100 mV s^{-1} . **b** The output voltage of MEG. **c** The self-discharge curve of the EC. **d** The leakage currents of the EC at various voltage windows. **e** The stable voltage output of the mp-SC for over 120 h at 90% RH and 25°C .

2. We all well know that the change of ambient temperature has a decisive impact on the evaporation of water inside a device, thereby affecting the overall performance of the device. The related research in this paper is relatively simple. I strongly recommend supplementing the corresponding studies and experiments, and adding the results and graphs to the main text.

Response: We thank for the reviewer’s detailed suggestion. As suggested, the impact of ambient temperature has been included in either the revised manuscript or the Supplementary Information file. Experimental results reveal that with increasing temperature from -18°C to 25°C , a significant enhancement in the MEG and mp-SC output current occurs (Fig. R2a–b), which could be induced by the ion migration rate increasement under higer tempreature (Membranes 2023, 13, 725; Chem. Rev. 2017, 117, 4759). However, after the device temperature rises to 60°C , the output voltage gradually decreases. This phenomenon may be attributed to the rapid migration of ions within the polyelectrolytes film due to the temperature increase, leading to a reduction in the ion concentration gradient and subsequently diminishing the output voltage. Numerical simulation results also validate a similar correlation between the device's output voltage and ambient temperature above mentioned (Fig. R2c–d).

Fig. R2 **a** The output voltage and current densities of the polyelectrolyte MEG at different temperatures. **b** The output voltage of mp-SC at different temperatures. **c** Numerical simulated model of the polyelectrolyte film. **d** The simulation results depict how the distribution of the electric field within the polyelectrolyte film varies with changes in temperature.

3. The authors mentioned in the text that “The mp-SC maintains outstanding cycling stability after 15 cycles, demonstrating stable and durable output performance”. However, they also said in the abstract and text that “this mp-SC demonstrates ~100% performance retention under bending for 1000 cycles”, and “the integrated device demonstrates a good capacitance retention rate of ~100% under bending states”. Whether the two statements are contradictory each other?

Response: We sincerely apologize for the inappropriate statements in the manuscript. In Fig. R3a, we evaluate the cycle stability of the mp-SC by conducting more than 15 cycles of moisture-charging/discharging curves under approximately 90% RH at 25°C. After the mp-SC has discharged its stored energy, it can spontaneously absorb moisture from the environment again, thereby returning to approximately 0.9 V. These results demonstrate the durable moisture-charging/discharging output performance of the mp-SC. In Fig. R3b, the output voltage of the mp-SC was recorded under cyclic mechanical bending and flat states. The mp-SC demonstrates a good voltage retention rate of ~100% after 1000 bending test. We have revised the related description in the revised manuscript.

Fig. R3 a Cycling stability of the mp-SC at 90% RH. b The voltage retention rate of the mp-SC after cyclic bending test.

4. In Figure 4, most of the graphs most have incorrect title numbers. Please revise carefully.

Response: We appreciate the reviewer's thorough review, and we have corrected this error in the revised manuscript. Additionally, we have carefully checked for any other potential errors throughout the manuscript.

Fig. R4 The performance of the mp-SC. a Schematic of the charging process of the mp-SC. b The voltage output of the mp-SC at different RH. c The voltage output of the mp-SC at different temperature. d Galvanostatic charge-discharge profiles of the mp-SC at different current densities. e The areal capacitance of the mp-SC at different current densities. f Comparison of the capacitance of the mp-SC with the recently reported ECs. Output voltage and current (g) as well as the power density (h) of the mp-SC with different electric resistors. i Comparison of the power density of the mp-SC and the recently reported moisture power generators. j Cycling stability of the mp-SC at 90% RH. k Optical images of the mp-SC at the bending state. l The voltage output

of the device at different bending states.

5. In addition, I suggest the authors to appropriately reduce the references that are their own works.

Response: As suggested, we have have removed some references of our own works.

Reviewer #2 (Remarks to the Author):

This manuscript reported on the development of an interesting flexible moisture-powered supercapacitor (mp-SC), which can absorb water from air and then self-charge electricity spontaneously. This is a wonderful experiment and the novel device is well-designed. The attractive power generation performance, ultra-long term energy storage ability, remarkable mechanical flexibility and large-scale integration capacity highlights the potential of mp-SCs as brand-new self-powered sources for wearable and flexible electronics. The study is of high quality and I strongly suggest considering this work for publications in Nature Communications after addressing some minor concerns as listed below:

Response: We really appreciate for the reviewer's valuable comment.

1. It is not clear what the carbon electrode is and how it looks like after blade coating. Please supplement more information.

Response: We thank the reviewer for this suggestion. In this work, a conductive carbon paste (CH-8, Jelcon Corp., Japan) and conductive carbon tape (No. 7321, Nisshin EM Corp., Japan) were adopted to serve as the electrode of the mp-SC. The conductive carbon paste is suitable for blade coating or screen printing to fabricate the integrated device and the large-scale flexible device array. The carbon paste shown good conductive after drying, which was previously used as the electrode of moist-electric generator (Energy Environ. Sci., 2019, 12, 972). As depicted in Fig. R5a-c, the carbon electrodes exhibit a smooth surface morphology. The related research have been added in the revised supplementary information.

Fig. R5 a The optic photo of the blade coating carbon electrode. SEM images of the

carbon electrode (b-c).

2.The authors should consider providing additional discussions on existing self-powered supercapacitors in the introduction section.

Response: According to the reviewer's recommendation, we have enriched the introduction section by including relevant research on self-charging capacitors at page 2 in the revised manuscript. “The development of self-charging integrated devices across one-dimensional fibers, two-dimensional films, three-dimensional bulk structures, and textile forms has emerged for various applications including health monitoring bioelectronics, sensors, and wearable electronics. Nevertheless, these self-charging processes are inherently intermittent, requiring external mechanical stimuli or specific geographic and climatic conditions.”

3.How about the thickness of this flexible mp-SC?

Response: The mp-SC is integrated by a polyelectrolyte moist-electric generator and a graphene based microsupercapacitor, the thickness of the mp-SC is ~180 μm . The thickness of the mp-SC is shown in Methods section in the revised manuscript.

4.The scale bars in Figure 3b and currently missing from the manuscript.

Response: Thanks for your suggestion, the Figure at page 8 in the revised manuscript have been updated.

5.In Figure 4i, the authors compared the performance with previously reported moist-electric generators. However, given the abundance of existing reports on moist-electric generators, the authors should provide a more detailed performance comparison, including materials, device structures, and other relevant factors.

Response: Thanks for your suggestion, we added the comparison of the mp-SC and the previously reported moist-electric generators. The detailed performance comparison, including materials, device structures, and other relevant factors are added in the revised manuscript and the revised supplementary information.

Table R1 Comparison of the power density of the mp-SC and the recently reported moisture power generators.

Materials	Device structures	Electrodes	Voltage (V)	Power density ($\mu\text{W cm}^{-2}$)
GO film	Thin film	Ag	0.7	27
PSS	Thin film	steel@Au	0.8	17
GON	Thin film	Al	0.04	12

PSS+PVA	Thin film	Ag NWs	0.6	7.9
PSS/PVA	Thin film	carbon tape	0.95	5.5
TiO ₂	Nanowire networks	Ag NW	0.50	4
PPy	Foam	Au	0.06	0.69
g-GOF	Film	Au	0.04	0.42
IPMEG	Film	rGO	0.18	0.1
GO/PAAS	Bulk	Au/Ag	0.60	0.07
a-GOM	Foam	Au	0.45	0.0184
Cellulose	Nanofibrous foam	Pt	0.11	3E-4
This work	Thin film	Carbon	0.9	49.38

6. In Figure 5h, it is not explicitly stated whether the thermohygrometer directly transmits the data to a smartphone through Bluetooth or if any relay device is involved in the process.

Response: In the experimental setup, the signal from the thermohygrometer is first transmitted via Bluetooth to a gateway device compatible with the temperature and humidity meter, and then transmitted from the gateway device to the smartphone.

Reviewer #3 (Remarks to the Author):

The author presents the novel configuration of a moisture-based nanogenerator and supercapacitor. Overall, the manuscript is well-written, but the current manuscript should be considered after addressing the following major questions.

Response: We express our gratitude to the reviewer for insightful recommendations to enhance the quality of our manuscript.

1. The title of the manuscript seems too broad. It should clearly describe the novelty of the work instead of the “review article”-like title.

Response: We appreciate the reviewer’s insightful suggestions. As suggested, we have changed the title of the article from "Moisture-powered supercapacitor for ultra-long term voltage stability" to "Moisture-enabled self-charging and voltage self-stabilizing supercapacitor".

2. The overall configuration should be reviewed. The author should explain the stacked structure of the supercapacitor and mp-SC more systematically. For better understanding, it is recommended to show the stacked structure of mp-SC in Figure A

with well-defined connections between SC and MEG. For example, in Figures 1f and 1g, it is quite difficult to understand, how the electrodes are connected to Supercapacitor.

Fig. R6 Schematic of the mp-SC. **a** Photos of the bilayer polyelectrolytes film and the rGO microelectrodes array obtained by direct laser writing as well as the scheme of the mp-SC. The mp-SC consists of a polyelectrolyte-based MEG and a graphene EC. **b-e** Schematic illustration of the fabrication process of mp-SC. **f** The schematic diagram of connection between between SC and MEG of the mp-SC.

Response: Thanks for your suggestion. In the mp-SC, the bottom electrode of the MEG is connected to one electrode of the interdigitated EC, while the top electrode is connected to the counter electrode of the EC. The detailed fabrication process of the mp-SC is provided in the Methods section in the revised manuscript. Besides, we have updated the schematic diagram of the mp-SC in the revised manuscript.

3. Also, the energy harvesting mechanism and energy storage from the moisture should be more clearly depicted as a schematic. In the current manuscript, it is too hard to understand the energy harvesting mechanism. Clearer schematic as well as description are required.

Response: By integrating the energy generation part and energy storage part with well-designed electrodes as indicated in Fig. R7, this mp-SC absorbs water from the air and

gradually dissociates and releases moveable H^+ and Cl^- at the MEG part (Fig R7a-b). Then, the asymmetrically distributed migratable ions (H^+ and Cl^-) in the bilayer polyelectrolyte film will diffuse to the opposite side driven by the concentration difference above mentioned and produce electrical power (Fig R7c). The voltage generated by the polyelectrolyte film is applied the carbon electrodes, and then acts on the rGO microelectrodes of the EC through the internally connected electrodes. Under the applied voltage bias, the electrolyte within the EC selectively adsorbs onto the surface of the rGO electrodes, facilitating the conversion of electrical energy into chemical energy (Fig R7d). Based on this method, the power energy produced by the MEG is stored in the EC via the electric double-layer mechanism. The related research have been added in the revised supplementary information.

Fig. R7 Schematic of the charging process of the mp-SC. **a** The initial state of the mp-SC. **b** The spontaneous water absorption and ion transport processes. **c-d** The ions in electrolyte within the EC selectively adsorbs onto the surface of the rGO electrodes under the applied voltage bias.

4. The author should show the video for real time generation of the electricity from the moisture-environment. For example, in real time, continuous changing environment from dry state to humid state with generation of electricity in oscilloscope would be enough.

Response: As suggestion, we have included videos showing the transition process of MEG and the mp-SC from low humidity (~5% RH) to high humidity (~90% RH) as show in Fig. R8. The videos have been uploaded as supplementary materials.

Fig. R8 The voltage test process of the MEG at 5% RH (a) and 95% RH (b). The voltage test process of the mp-SC at 5% RH (c) and 95% RH (d).

5. The author should explain the different parameters in electrochemical impedance spectroscopy of supercapacitors i.e. charge transfer resistance, and series resistance under the different bending conditions. Also, the strong connection between the charge transfer resistance and charging time when connected with MEG should be explained in detail. The EIS part at the higher frequency region should be present in the inset of Figure 2g.

Response: As suggested, a schematic of the equivalent circuit model is added in the revised supplementary information. The equivalent series resistance of EC is composed of three parts (Fig. R9a), including intrinsic ohmic resistance (R_{Ω} , $302 \Omega \text{ cm}^2$), interfacial charge transfer resistance (R_{CT} , $24 \Omega \text{ cm}^2$) and Warburg diffusion resistance (R_W , $160 \Omega \text{ cm}^2$). Under different bending conditions, the impedance of EC exhibits negligible variation, demonstrating the exceptional flexibility (Table R2). Besides, MEG process steady electric power output by spontaneous water absorption (Fig. R9b). Therefore, the charging process should also be stable based on the stable internal resistance of EC and stable charging voltage of MEG in mp-SC, demonstrating a good voltage retention rate of $\sim 100\%$ under bending and flat states after 1000 bending cycles (Fig. R9c-d). In addition, the EIS part at the higher frequency region has been added in the revised manuscript as suggested.

Fig. R9 **a** The equivalent circuit of the EC. **b** Output voltage of the MEG. **c** The voltage output of the mp-SC at different bending states. **d** The voltage retention rate of the mp-SC after cyclic bending test.

Table R2 The equivalent series resistance change of the supercapacitor under different bending conditions.

Bending angle	intrinsic ohmic resistance ($\Omega \text{ cm}^2$)	interfacial charge transfer resistance ($\Omega \text{ cm}^2$)	Warburg diffusion resistance ($\Omega \text{ cm}^2$)
Flat	302	24	160
90°	297	17	156
180°	307	26	164

6. In general, the fabricated supercapacitor lacks novelty in terms of material as rGO has already reported multiple times.

Response: Indeed, rGO is widely employed in the fabrication of supercapacitors (Science, 2012, 335, 1326; Nature Communications, 2013, 4, 1475). In this study, we employed laser direct writing strategy for supercapacitor fabrication based on the excellent electrochemical properties and mechanical flexibility, facilitating convenient integration with different parts and large-scale production of the mp-SC. Future investigations can further explore pseudo-capacitive materials for energy storage devices to achieve enhanced energy storage capacity, depending on the development of appropriate mechanical charectors and fabrication processes.

7. It is commonly observed that potential drops if the supercapacitor is connected

without any rectifier. The author should report the potential drops with different levels of humidity level once the supercapacitor is fully charged. How does the author balance the charge flow or overcome the reverse flow of current from the supercapacitor to MP. The reverse flow can again neutralize the charge distribution of H^+ and Cl^- . Please explain the mechanism in detail.

Response: Rectifiers are widely employed in pulsed electricity-generating devices for energy storage applications, such as the integration of piezoelectric or triboelectric generators with supercapacitors (Adv. Mater. 2020, 32, 2002180). In such scenarios, where generator devices produce short-duration and high-frequency current pulses, a rectifier is necessary to convert the generated alternating current into direct current for storage in the capacitor. However, the generator utilized in this study demonstrates a stable output characteristic akin to constant energy output. Thus, the energy generated in single cycle is sufficient to charge the EC to its operational voltage. When the leakage current of inner EC balances with charging current of inner MEG induced by moisture, the output voltage of the mp-SC reaches a stable value. Based on this moisture-enabled charging process and the phenomenon of open circuit self-discharge is inevitable in EC, the final voltages of mp-SC will be little lower than that of individual MEG under the same humidity value (Fig. R10 a-b). For example, the output voltage of the mp-SC reaches a stable value of about 0.9 V (Fig. R10c) at 90% RH, while individual MEG demonstrates a sustained output voltage of 1.27 V under the same condition.

Fig. R10 a The voltage and current density output of the MEG at different RH. **b** The voltage output of the mp-SC at different RH. **c** The stable voltage output of the mp-SC for over 120 h at 90% RH and 25°C.

8. The author should explain the detailed mechanism that how functional groups in the bipolar layer can lead to absorbing and desorbing water.

Response: The polar sulfonic acid functional groups in polystyrene sulfonic acid and the amino functional groups in polydiallyl dimethyl ammonium chloride confer excellent water absorption properties to the polyelectrolytes film (J. Mol. Struct. 2001, 595, 111). These functional groups can form hydrogen bonds with water molecules, enhancing the surface polarity (Fig. R11a). This imparts excellent wettability to the film and promotes the adsorption of water molecules. After reducing the moisture content in ambient environment, while excessive absorbed water can desorb from the polyelectrolytes film dependent the ambient RH as shown in Fig. R11b.

Fig. R11 **a** Contact angle photos of PDDA and PSS+PVA membranes. **b** The mass change of the polyelectrolytes film at different RH.

9. The author should explain the optimization of thickness for PSS and PDDA. The effect of surface potential on the thickness of PSS and PDDA should be addressed in detailed.

Response: Following the reviewer's recommendation, we conducted numerical simulation and experiment tests to assess the moisture generation performance of polyelectrolytes films with varying PSS and PDDA ratio or film thicknesses (25–400 μm). As shown in Fig. R12a, for a total thickness of 100 μm in the film, the highest output voltage is achieved when the ratio of PDDA to PSS thickness is 3:7 according to the numerical simulation results. Moreover, we tested the output performance of polyelectrolytes films with varying thicknesses. As shown in Fig. R12b, as the thickness of the polyelectrolytes film increased from 25 μm to 100 μm , the voltage output increase from 0.86 V to 1.27 V. However, further increasing the thickness did not lead to a higher output voltage for the device. This phenomenon may arise from that as the thickness of the membrane increases, a larger quantity of ions becomes accessible, enabling a higher current output. Nonetheless, with further increases in membrane thickness, the elongated diffusion path for ions leads to diminished conductivity, initially resulting in an increase followed by a decrease in the output current from 2.7 $\mu\text{A cm}^{-2}$ to 1.21 $\mu\text{A cm}^{-2}$.

Fig. R12 a The voltage output of the polyelectrolytes film with different PSS and PDDA thick ratio. **b** Voltage and current density of the moist-electric generator at different thicknes.

10. The captions in Figure (4) are not well matched with the figure number. Please correct it.

Response: We appreciate the reviewer's thorough review, and we have corrected this error. Additionally, we have carefully checked for other potential errors throughout the manuscript.

Above all, the manuscript has been revised carefully as suggested. Thanks very much for comments from all reviewers again.

REVIEWERS' COMMENTS

Reviewer #1 (Remarks to the Author):

I have read all the replies the author answered in the review report. The author addressed all my concerns in great detail, and the revised manuscript is suitable for the publication. I suggest the publication of this manuscript at the current version.

Reviewer #2 (Remarks to the Author):

Authors have addressed comments from reviewers well. This work about moisture enabled self-charging and voltage-stable supercapacitor demonstrates promising power generation and storage ability in air, which will attracted much attentions by researchers focusing on energy-related studies.

Reviewer #3 (Remarks to the Author):

The revised manuscript is ready for publication.

Reviewer #1 (Remarks to the Author):

I have read all the replies the author answered in the review report. The author addressed all my concerns in great detail, and the revised manuscript is suitable for the publication. I suggest the publication of this manuscript at the current version.

Response: We appreciate the suggestions provided by the reviewer for revising the manuscript, and we are grateful for their efforts.

Reviewer #2 (Remarks to the Author):

Authors have addressed comments from reviewers well. This work about moisture enabled self-charging and voltage-stable supercapacitor demonstrates promising power generation and storage ability in air, which will attract much attention by researchers focusing on energy-related studies.

Response: We appreciate the positive feedback from the reviewer and sincerely thank for the valuable assistance during the review process.

Reviewer #3 (Remarks to the Author):

The revised manuscript is ready for publication.

Response: We appreciate the reviewers' evaluations and valuable suggestions for revising the manuscript, which have helped meet the requirements of the journal.